# Machine learning analysis of the UK Biobank reveals IGF-1 and inflammatory biomarkers predict Parkinson's disease risk

**Michael Allwright**[1]*, **Hamish Mundell**[2], **Greg Sutherland**[2], **Paul Austin**[1☉], **Boris Guennewig**[1☉]

1 Brain and Mind Centre and School of Medical Sciences, Faculty of Medicine and Health, The University of Sydney, Camperdown, NSW, Australia, 2 Charles Perkins Centre and School of Medical Sciences, Faculty of Medicine and Health, University of Sydney, Camperdown, NSW, Australia

☉ These authors contributed equally to this work.
* michael.allwright@sydney.edu.au

**Data Availability Statement:** All relevant data are within the manuscript and its Supporting Information files. The individual patient level data is

## Abstract

### Introduction

Parkinson's disease (PD) is the most common movement disorder, and its prevalence is increasing rapidly worldwide with an ageing population. The UK Biobank is the world's largest and most comprehensive longitudinal study of ageing community volunteers. The cause of the common form of PD is multifactorial, but the degree of causal heterogeneity among patients or the relative importance of one risk factor over another is unclear. This is a major impediment to the discovery of disease-modifying therapies.

### Methods

We used an integrated machine learning algorithm (IDEARS) to explore the relative effects of 1,753 measured non-genetic variables in 334,062 eligible UK Biobank participants, including 2,719 who had developed PD since their recruitment into the study.

### Results

Male gender was the highest-ranked risk factor, followed by elevated serum insulin-like growth factor 1 (IGF-1), lymphocyte count, and neutrophil/lymphocyte ratio. A group of factors aligned with the symptoms of frailty also ranked highly. IGF-1 and neutrophil/lymphocyte ratio were also elevated in both sexes before PD diagnosis and at the point of diagnosis.

### Discussion

The use of machine learning with the UK Biobank provides the best opportunity to explore the multidimensional nature of PD. Our results suggest that novel risk biomarkers, including elevated IGF-1 and NLR, may play a role in, or are indicative of PD pathomechanisms. In particular, our results are consistent with PD being a central manifestation of a systemic

held by UK Biobank and would need their approval for an independent researcher to access.

**Funding:** The authors received no specific funding for this work.

**Competing interests:** The authors have declared that no competing interests exist.

inflammatory disease. These biomarkers may be used clinically to predict future PD risk, improve early diagnosis and provide new therapeutic avenues.

## Introduction

Parkinson's Disease (PD) is the second most common neurodegenerative disease affecting over 6 million people worldwide, and has seen a 3-fold increase in the last 30 years [1]. It is a movement disorder associated with a high level of disability for individual sufferers and a significant burden for caregivers. To prospectively screen for PD and better elucidate the disease mechanism, there is a need to identify blood-borne biomarkers, as well as environmental and genetic factors that are associated with greater risk or are protective. This is critical because neurodegenerative processes in dopamine neurons of the midbrain start many years before PD diagnosis. Thus, there is a need to identify future risks, enabling early interventions to be offered, which may take the form of beneficial lifestyle changes or the development of novel neuroprotective agents.

Exposure to pesticides, consumption of dairy products, melanoma and traumatic brain injury are thought to increase the likelihood of PD diagnosis [2], while smoking, caffeine intake, high serum and urate concentrations, physical activity and the use of ibuprofen and other medications are considered protective [3, 4]. There has also been some recent interest in the link between increased insulin like growth factor (IGF-1) and inflammation in the early phase of PD [5–7], whilst higher cholesterol levels are thought to cause a reduction in PD risk [8]. However, most studies that have examined PD risk to date consist of univariate hypothesis tests controlling for confounder variables of known risk factors. There have been few studies considering the associations from a wide range of variables together without *a priori* assumption. Employing machine learning to consider a complete set of candidate risk factors enables the significance of both established and novel risk factors to be evaluated in an unbiased manner and, significantly, does not require any prior knowledge of PD risk factors.

The UK Biobank (UKB) is the largest deeply phenotyped epidemiological study globally. A study has looked at the interaction between genetics and established risk factors in predicting PD [9]. A separate study has confirmed the importance of anxiety, depression, family history of PD, excessive daytime sleepiness, pesticide exposure and being underweight, using logistic regression [10]. The association between lower lymphocyte count and PD has also been demonstrated in the UKB [11]. The methods applied in these studies were limited to logistic regression, Cox Proportional Hazards Survival Analysis and univariate approaches controlling for known risk factors. While these determine individual odds or hazard ratios associated with a small set of predictor variables, they neglect interaction and non-linear effects between variables. They cannot model more than a handful of variables. They, therefore, cannot take full advantage of the breadth of studies like the UKB. Cutting-edge machine learning techniques have already been applied to the UKB to determine risk factors for cardiovascular disease [12]. The ADNI dataset has been used with gradient boosting and SHAP to model APOE4 [13], and neuroimaging data in those with mild cognitive impairment (MCI) [14].

The Integrated Disease Explanation and Risk Scoring platform (IDEARS) is an automated data processing, machine learning and visualisation platform that combines hospital inpatient data, clinical assays and questionnaire data and applies feature engineering, classification models and feature importance methodologies to develop; an automated risk score and model performance metrics; a ranking of variables with the most significant associations with a disease

using the novel kernel SHAP methodology to infer feature importance [14, 15] and; visualization of disease risk profiles for specific variables. This study applies the IDEARS platform to determine individual PD risk for those aged 50–70. It validates this by comparing it to a model derived from well-established risk factors from previous studies. Through this process, we present a novel ordered set of PD risk factors for consideration, split by gender.

## Materials and methods

### Ethics approval

UK Biobank has approval from the North West Multi-centre Research Ethics Committee (MREC) as a Research Tissue Bank (RTB) approval. This approval means that researchers do not require separate ethical clearance and can operate under the RTB approval. All methods in this study were performed in accordance with the relevant guidelines and regulations of MREC. Participants signed an informed consent with the UKB for the data to be accessed by external researchers in order to undertake research studies such as this. All UKB data used in this study was fully anonymised and non-identifable prior to us gaining access.

### Sample selection

The UKB study recruited 502,253 subjects aged 37–73 years in the United Kingdom between 2006–2010, performing a raft of clinical measurements and assays, including clinical pathology screens, genotyping, neuroimaging and cognitive testing, as well as medical information, health records and self-reported demographic and wellness data [16]. "Baseline" was defined as the first date each participant attended a UKB assessment centre. At that point, the data relating to predictor variables for each participant in the study were collected (Fig 1).

To establish our cohort, we defined those diagnosed with Parkinson's as having ICD10 code G20 recorded in their hospital inpatient records, dated 2 or more years after baseline as "PD cases". We excluded those who died within 10 years of baseline of something other than PD and those younger than 50 or older than 70 at baseline, due to the much lower risk of idiopathic PD before age 50 and the small number of participants over 70. For the main analysis, we further excluded those participants who were already diagnosed with PD at baseline, based

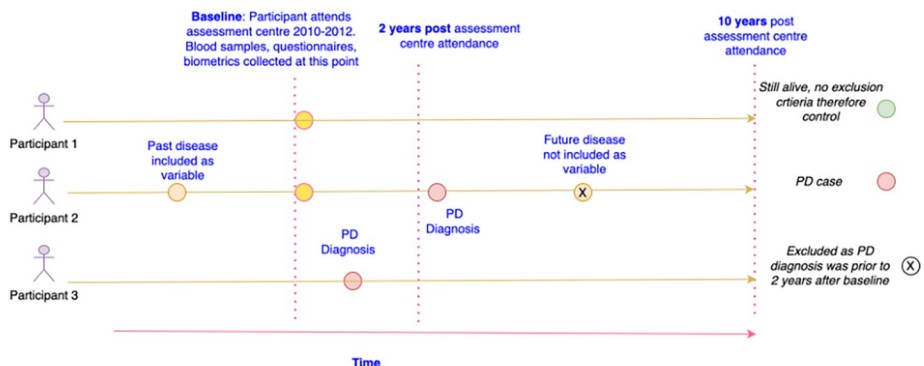

**Fig 1. UK Biobank timeline and case/control inclusion/exclusion criteria.** In this case participant 1 would be selected in our analysis as they attended the assessment centre and did not fulfil any exclusion criteria, nor were they ever diagnosed with PD. Participant 2 is included as a case, with a past disease coded as an independent variable as it was diagnosed before their attendance at the assessment centre, and they were diagnosed with PD over 2 years after their attendance. Participant 3 is excluded as they received a PD diagnosis before attending the 2 years post baseline assessment centre.

either on an ICD10 code or the UKB derived data field "date of parkinson's disease report", or who developed PD within 2 years of baseline to avoid skewing the predictions with the immediate pre-symptomatic phase of the disease. This left 336,781 participants in the cohort, 2,719 of whom developed PD within an average of 8.0 years from baseline. For the stratification, by gender analysis, the same data was considered but in this case all individuals who were assigned the G20 ICD10 code at any point between 5 years before first attendance at the assessment centre and 10 years after attendance were also included.

## IDEARS platform

The IDEARS platform was developed by the authors of this paper (Fig 2). IDEARS applies machine learning to health-related questionnaire data, longitudinal inpatient data (ICD10), blood assays, genetic and neuroimaging data. It can be applied to any disease categorized by a set of ICD10 codes and with a large enough number of cases (typically n>500). The complete codebase can be accessed at *https://github.com/binfnstats/idears_orig*.

## Data integration layer (data processing and feature engineering)

The most recent inpatient data received (release date: September 2021) was used to identify the complete set of ICD10 codes corresponding to any condition for which the number of cases across our cohort exceeded 200 and a diagnosis was given before baseline. This consists of the distinct primary/main diagnosis codes recorded for each participant across all their hospital inpatient records. This resulted in 1,101 binary features corresponding to a participant having (1) or not having (0) a given disease at baseline. We derived a variable for the total number of conditions each participant had at baseline as well as the total number of conditions within 20 illness groups defined through the ICD10 package in python.

Blood assay, clinical and self-reported questionnaire data were merged, and all variables which had greater than 80% non-missing observations were selected for the subsequent analysis. Feature engineering–which involved one-hot encoding and conversion of variables with a natural order to a numeric ordinal score–was performed as part of the IDEARs platform.

A set of variables to represent risk factors with known associations to PD were developed; these included age, gender, neuroticism score, constipation, coffee intake, smoking status, exposure to pesticides, urban/rural living, depression, level of activity, a family history of PD, use at baseline of beta-blockers, ibuprofen and non-steroidal anti-inflammatories.

## Risk scoring and model explanation layer

Two variable sets were used for modelling–$V_A$ which included all variables selected through the data integration layer (total variables = 1,753) and $V_C$ which included only the consensus

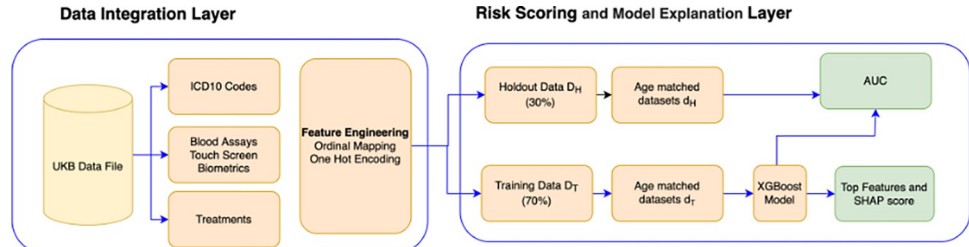

**Fig 2. Integrated disease explanation and risk scoring platform (IDEARs).** An automated platform to facilitate the data integration of large health datasets, transforming data fields into useful features, performing risk scoring, and determining ranked feature importance for any disease.

of risk factors derived from high-quality meta-analyses [3, 4]. To avoid data leakage, data was first split into a training dataset, $D_T$, and a holdout dataset, $D_H$, with $D_T$ containing a random sample of 80% of cases and 80% of controls, with the remaining 20% of each selected for $D_H$ and reserved for validation. Mean imputation of missing values was performed separately on both $D_T$ and $D_H$.

The full risk-scoring and model explanation layer was applied to $D_T$ with 100 resamples, thus enabling each control participant in $D_T$ to be used in the analysis at least 6 times.

The resampled data was prepared by: 1) Building 100 observational subsets $d_T \subset D_T$ which each consisted of all PD cases in $D_T$ and 40 age matched controls sampled from $D_T$ without replacement until all were used. 2) Further splitting each $d_T$ at random into a training dataset ($d_{Ttrain}$– 70% of records) and a testing dataset ($d_{Ttest}$– 30% of records); 3) Applying mean imputation of missing values on both $d_{Ttest}$ and $d_{Ttrain}$.

Three classification model types were applied to the first 10 $d_{Ttrain}$ with the AUC value calculated on the corresponding $d_{Ttest}$ for each iteration. XGBoost [17] (mean AUC = 0.67) outperformed Support Vector Machines (mean AUC = 0.66), random forest (mean AUC = 0.65) and logistic regression (mean AUC = 0.59) and was therefore selected for the subsequent analysis.

Hyper-parameter tuning was performed using a grid search with the following hyperparameters: learning rate, minimum child weight, maximum depth, and positive weighting scale which determine the structure of the XGBoost algorithm. The set of hyperparameters which generated the highest AUC values for the model were selected to define model M which was trained on each dataset in the subsequent analysis.

Model M was then trained on each $d_{Ttrain}$ and the mean SHAP [15] score was calculated on each corresponding $d_{Ttest}$. The variables with the top 50 total SHAP score across all 100 iterations were then selected for the subsequent analysis. The process was repeated for male and female subsets of $D_T$. All of these variables were put together for subsequent modelling, with the total number comprising 108 independent variables ($V_S$).

For male only, female only and a combined cohort, Model M was then further trained on each $d_{Ttrain}$ for the new candidate features $V_S$ alongside known associations $V_C$ ($M_{SC}$) and just the known associations $V_C$ ($M_C$). For $M_{SC}$ the mean SHAP score was calculated for each variable on the holdout dataset $D_H$. The AUC metric was calculated on the holdout dataset $D_H$ for trained models $M_{SC}$ and $M_C$ to determine the discriminative performance of the model with just known associations ($M_C$) with the model on known associations and all others ($M_{SC}$). AUC was sampled for each model iteration and an unpaired 2-sample t-test was used to evaluate whether there were statistical differences in mean AUCs between $M_{SC}$ and $M_C$. For a list of the variable sets $V_S$ and $V_C$ see S1 File.

## Stratification by gender and disease progression for key variables

The variables with the highest SHAP score in the above analysis were manually grouped into 'biometric', 'blood biomarkers', 'cardiovascular', 'demographic', 'frailty' and 'inflammation' categories. For the variables in each group, we examined the data and reintroduced those who already had been diagnosed with PD at baseline to determine the difference in value of each variable when summarised by "Years of PD". This variable was calculated for each participant as the date at which the participant attended the assessment centre (baseline) minus the date at which they had the PD ICD10 code reported. A "No PD" group was formed of participants that never had a PD ICD10 code reported. For both the male and female cohorts unpaired 2-sample t-tests were performed to compare the means for each variable in the non-PD group with the means of each variable at each disease stage: 5–10 years before disease diagnosis, 0–5

years before disease diagnosis and 0–5 years post disease diagnosis respectively. A Benjami-Hochberg correction [18] was performed on the resulting p values to correct for multiple comparisons. All p values presented are the adjusted p values.

## Results

A cohort of 334,062 UKB participants met the inclusion criteria, of which 2,719 received a clinical diagnosis of Parkinson's disease during the observation period (Table 1).

The IDEARS model, applied to $V_A$ (the full set of variables) had the best performance (Mean AUC = 0.672), compared to $V_C$, a comprehensive set of known risk factors identified from several high-quality meta-analyses (0.620). The AUROC graphs demonstrate superior performance of the IDEARS model in the total dataset, as well as when dividing the dataset by gender (Fig 3). A statistically significant performance advantage of the IDEARS model evaluated on holdout dataset $D_H$ is demonstrated based on 100 resamples of the training dataset compared to known associations (P<0.001, Fig 3A). The significant performance advantage is maintained with male and female datasets, with mean AUCs of 0.646 (P<0.001, Fig 3B) and 0.630 (P<0.001, Fig 3C), compared to 0.575 and 0.567 respectively.

### Top features

The top features from the IDEARS model are shown in Fig 4A. Variables from the meta-analyses which did not feature in our list of most important features, by average SHAP score, were smoking status, traumatic brain injury and caffeine consumption. Male gender was the feature with the highest mean SHAP score, in line with expectations given the known association with PD.

IGF-1 (3rd) stood out as a novel biomarker associated with an increased risk of PD. Whilst, features indicative of overall frailty were also associated with greater risk of PD, with a self-reported overall health rating from poor through to excellent (2nd), usual walking pace (4th), number of treatments/ medications taken (5th) and hand grip strength (Right, 27th). Features relating to inflammation were important, with increased neutrophil/lymphocyte ratio (NLR) (15th) and neutrophil percentage (18th) being associated with PD, and increased C-reactive protein (7th) and lymphocyte count (10th) being protective. Moreover, taking of non-steroidal anti-inflammatories was protective (21st). Cardiovascular and body fat variables appear to impact the risk of PD, with larger waist circumference (14th) being causative and elevated total cholesterol being protective (16th).

The top features from the IDEARS model for 1,698 males and 1,021 females are shown in Fig 4B & 4C, and the comparison of relative risk for each feature between males and females is

**Table 1. Population characteristics.**

|  | Controls | Cases | Total | Mean age at baseline | Mean time to diagnosis (years) | Mean Age at diagnosis |
|---|---|---|---|---|---|---|
| Male | 148,283 | 1,698 | 149,981 | 60.4+/-5.1 | 8.1+/-2.8 | 71.7+/-5.2 |
| Female | 185,779 | 1,021 | 186,800 | 60.0+/-5.1 | 8.0+/-2.9 | 71.2+/-5.2 |
| Total | **334,062** | **2,719** | **336,781** | **60.2+/-5.1** | **8.0+/-2.9** | **71.5+/-5.2** |
| White | 318,852 | 2,619 | 321,471 | 60.2+/-5.1 | 8.1+/-2.9 | 71.5+/-5.2 |
| Black | 3,713 | 22 | 3,735 | 58.3+/-5.5 | 8.3+/-2.6 | 71.3+/-5.3 |
| South Asian | 5,270 | 40 | 5,310 | 59.0+/-5.3 | 7.9+/-2.6 | 71.0+/-5.4 |
| Chinese | 832 | 1 | 833 | 58.0+/-4.9 | 12.0 | 69.0 |
| Mixed | 1,330 | 8 | 1,338 | 58.5+/-5.3 | 6.9+/-3.4 | 66.9+/-6.7 |
| Other | 4,065 | 29 | 4,094 | 59.2+/-5.2 | 7.8+/-3.2 | 71.1+/-6.1 |
| Total | **334,062** | **2,719** | **336,781** | **60.2+/-5.1** | **8.0+/-2.9** | **71.5+/-5.2** |

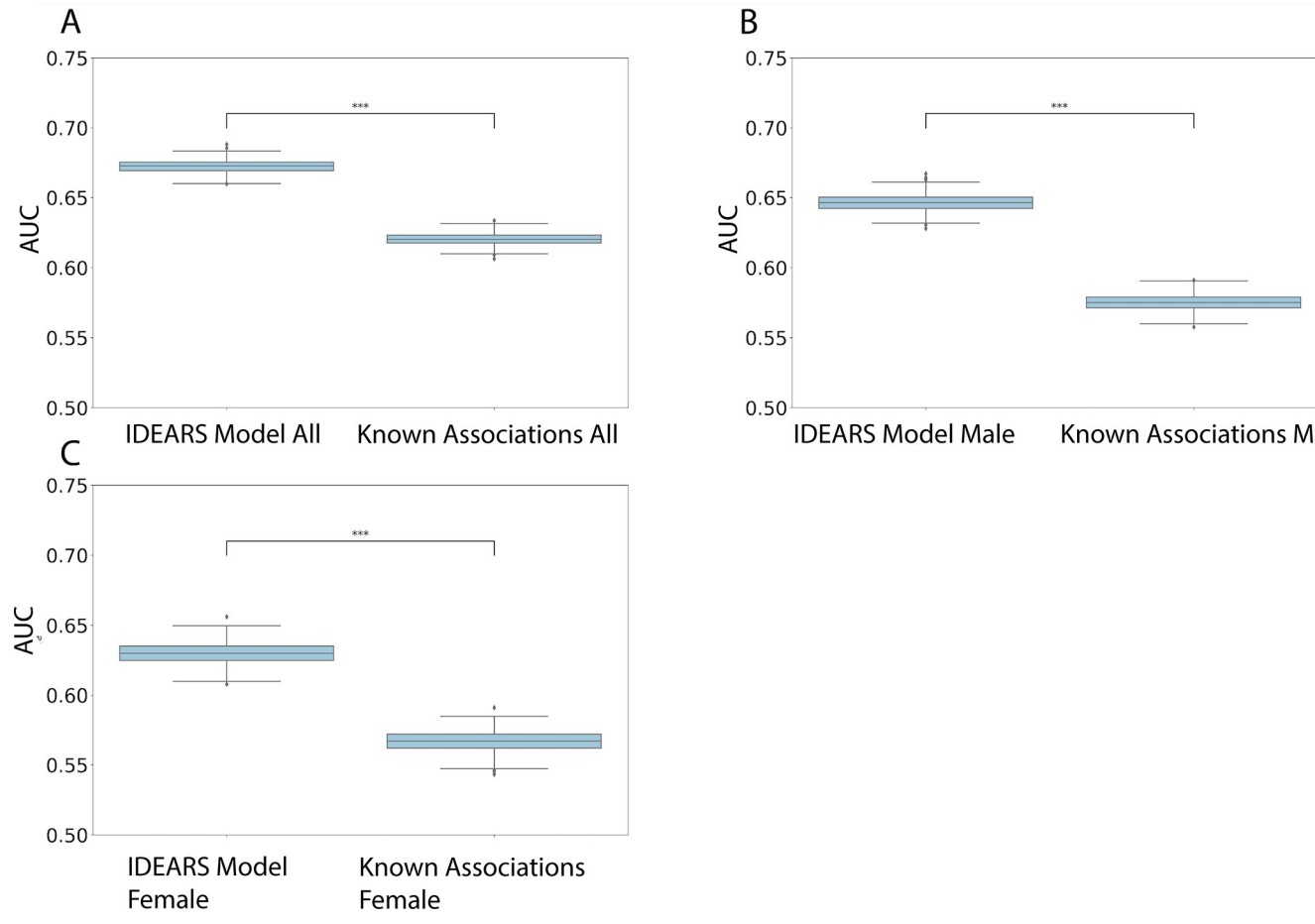

**Fig 3.** Boxplots of AUC results from 100 resamples on the combined set of features from the IDEARS model ($V_S$—set of features with highest mean SHAP score) compared to all known associations based on current meta-analysis ($V_C$) for A) the entire cohort, B) males and C) females. In each case, the mean IDEARS model AUC was statistically higher (***P<0.001) than the mean Known Associations model based on an unpaired 2-sample t-test.

shown in Fig 4D. The removal of gender as a feature in the model leads to a relative increase in some feature's importance and the appearance of some additional features in the gender segregated lists. In males elevated IGF-1 (1st) appeared to be a more important risk factor than in females (3rd). Urate (7th males and 9th females) and alanine aminotransferase (ALT) were protective in both sexes (17th males, 24th females), and frailty-related features were mostly of equal importance in both sexes. Neutrophil percentage (6th), NLR (9th) and having a parent with PD (20th) were more associated with PD in males, whilst cholesterol (12th) and triglycerides (5th) were more protective of PD, suggesting inflammation, cardiovascular and genetic factors may be more important in males. In females, vitamin D (5th), C-reactive protein (6th), glycated haemoglobin, (HbA1c), a marker of elevated blood sugar in the last 3 months, (13th) and forced vital capacity (18th) were protective, whilst chest pain or discomfort (7th), self-reported nervous feelings (8th) and bilirubin (12th) were associated with increased PD risk.

## IGF-1, AST:ALT ratio, hba1c, Urate and Creatine levels

Fig 5 shows IGF-1, AST:ALT ratio and HbA1c in the 10 years preceding and 5 after a PD diagnosis in males and females compared to the non-PD group. IGF-1 was significantly elevated at both -10 to -5 years (22.87 +/- 5.74, P<0.001) and -5 to 0 years before diagnosis (23.46 +/- 6.14,

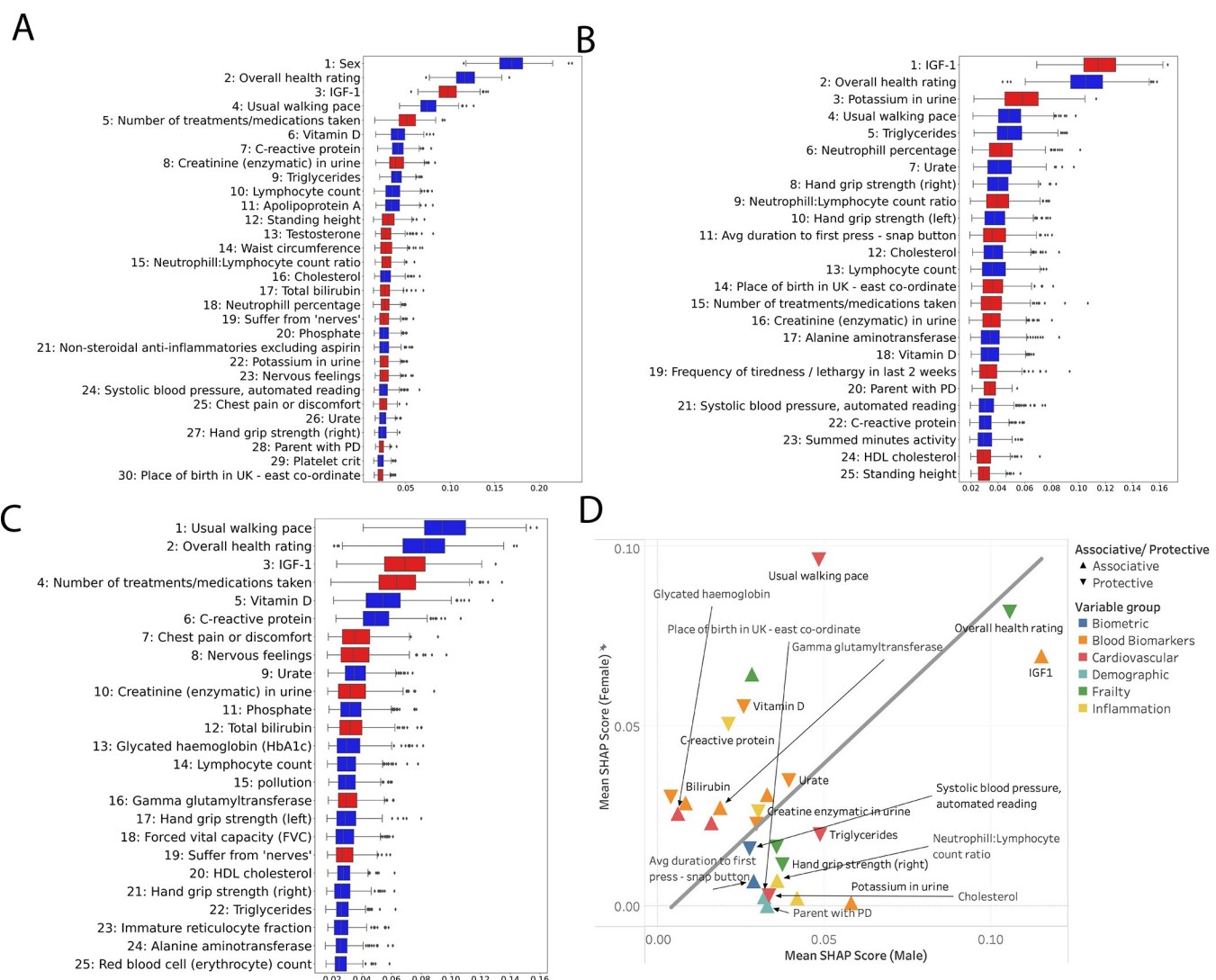

**Fig 4.** Box plots indicating the mean SHAP score of the top features from the IDEARS model for the entire cohort (**A**), males only (**B**) and female only (**C**). Those for which a higher value had a positive impact on the model output (i.e. making a PD diagnosis more likely) are coloured red. Those for which a higher value had a negative impact (making PD less likely) are coloured blue. (**D**) Scatter chart with SHAP feature importance for males versus females. Variables are colour coded in groups, and Δ denotes a causative association with PD, ∇ denotes a protective association.

P<0.001) and 0–5 years after diagnosis in males (24.78 +/- 7.68, P<0.001) compared to non-PD (21.47+/-5.33, Fig 5A). In females, IGF-1 was significantly elevated at both -10 to -5 years (21.04 +/- 5.86, P<0.001) and -5 to 0 years before diagnosis (20.89 +/- 6.38, P<0.05) and 0–5 years after diagnosis (21.87 +/- 6.27, P<0.01) compared to non-PD, (20.13 +/- 5.40, Fig 5B).

On the basis of ALT being protective in both sexes we investigated the common clinical indicator of liver disease, the AST:ALT ratio which was significantly elevated at both -10 to -5 years (1.20 +/- 0.40, P<0.05) and -5 to 0 years before diagnosis (1.29 +/- 0.41, P<0.001) and 0–5 years after diagnosis in males (1.39 +/- 0.55, P<0.001) compared to non-PD (1.17 +/- 0.39, Fig 5C). In females, the AST:ALT ratio was not significantly increased compared to non-PD in any of the breakdowns. Therefore, IGF-1 and the AST:ALT ratio are all elevated in males prior to diagnosis, whereas in females this only applies to IGF-1. According to the SHAP chart HbA1c showed a protective association with the risk of PD in females. However,

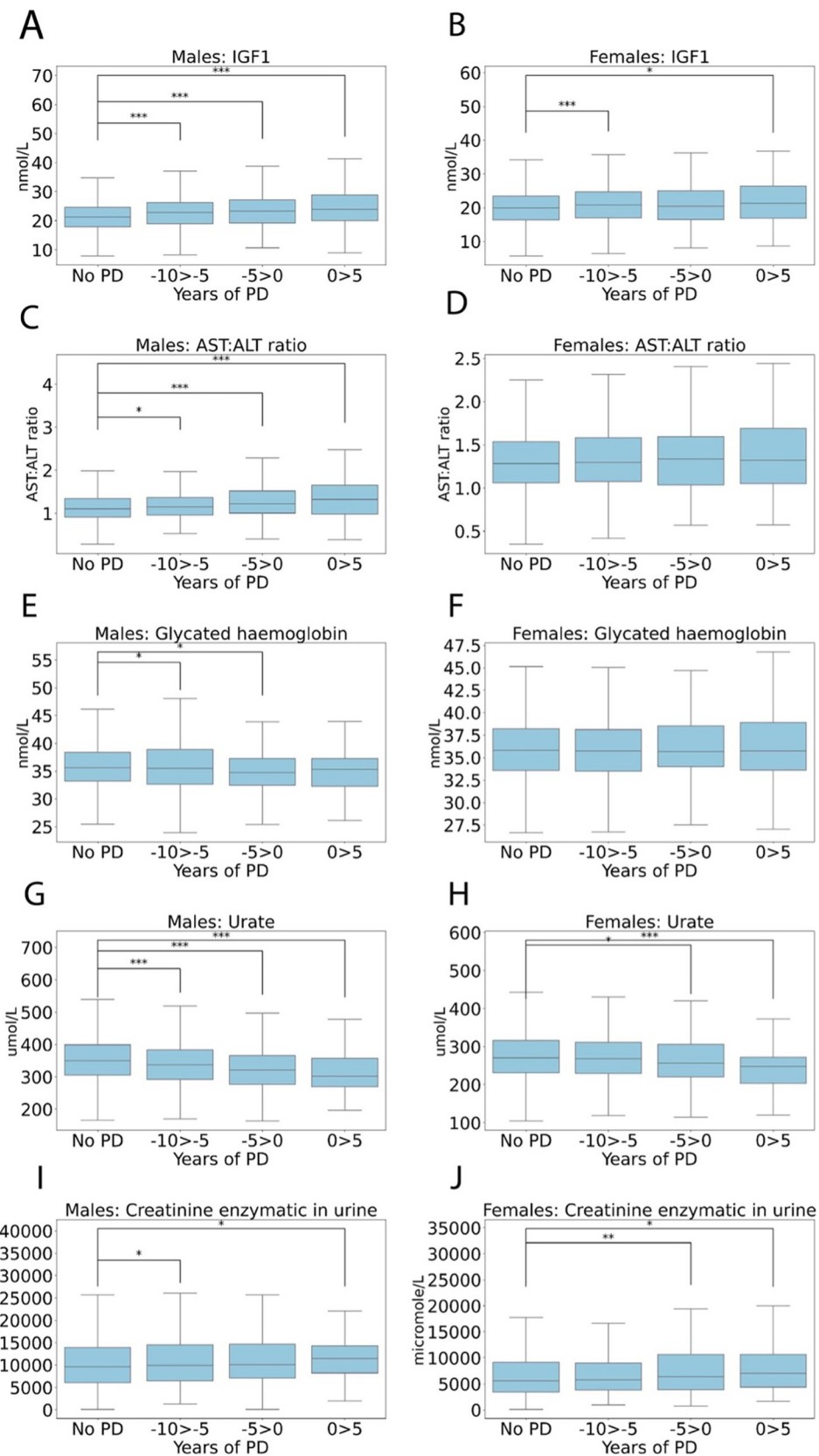

**Fig 5.** Box plots showing blood levels of IGF-1 levels, AST:ALT ratio, glycated haemoglobin, Urate and Creatinine for (**A, C, E, G, I**) males and females (**B, D, F, H, J**) in the 10 years preceding and 5 after a PD diagnosis in 2,036 males and 1,242 females compared to the non-PD group. AST: aspartate aminotransferase; ALT: Alanine transaminase. Mean +/- SD. An unpaired 2-sample t-test was used for statistical comparisons, with Benjami-Hochberg correction for multiple comparisons used to calculate the adjusted p values, *P<0.05, **P<0.01 and ***P<0.001.

looking at the values over time, HbA1c was significantly elevated in males 10 to 5 years before diagnosis (37.32 +/- 8.27, P<0.05) compared to non-PD (36.77 +/- 7.26), but was significantly reduced 5 to 0 years before diagnosis (35.88 +/- 6.85, P<0.05). For females there were no significant group differences.

Urate levels were significantly reduced at both -10 to -5 years (343.00 +/- 74.22, P<0.001), -5 to 0 years before diagnosis (326.51 +/- 67.86, P<0.001) and 0–5 years after diagnosis in males (320.91 +/- 78.97, P<0.001) compared to non-PD (354.63 +/- 70.96, Fig 5G) in males. In females, Urate levels were significantly reduced at both -5 to 0 years before diagnosis (266.11 +/- 71.64, P<0.05) and 0–5 years after diagnosis (250.14 +/- 75.82, P<0.001) compared to non-PD (274.48 +/- 67.69, Fig 5H). Creatinine (enzymatic) levels in urine were significantly elevated at both -10 to -5 prior to diagnosis (11133.64 +/- 6211.47, P<0.05 and 0–5 years after diagnosis in males (11956.21 +/- 5857.89, P<0.05) compared to non-PD (10600.11 +/- 5892.15, Fig 5I) in males. In females, creatinine levels were significantly elevated at both -5 to 0 years before diagnosis (7884.78 +/- 5547.44, P<0.01) and 0–5 years after diagnosis (8102.24 +/- 4592.67, P<0.05) compared to non-PD (6885.28 +/- 4669.08, Fig 5J).

Fig 6 shows inflammatory variables in the 10 years preceding and 5 after a PD diagnosis in males and females compared to the non-PD group. Neutrophil count was significantly elevated -10 to -5 years (4.38 +/- 1.36, P<0.05), 0–5 years prior to diagnosis (4.53 +/- 1.45, P<0.001) compared to non-PD in males (4.27 +/- 1.38, Fig 6A). In females, neutrophil count was significantly elevated 0 to 5 years after diagnosis (4.60 +/- 1.44, P<0.01) compared to non-PD (4.10 +/- 1.32). Lymphocyte count was significantly reduced at 10–5 years before diagnosis (1.80 +/- 1.31, P< 0.05), 0–5 years prior to diagnosis (1.69 +/- 0.63, P<0.01) and 0–5 years after diagnosis (1.71 +/-0.68, P<0.01) in males compared to non-PD (1.90 +/- 1.31, Fig 6C). In females, lymphocyte count was significantly decreased 0–5 years after diagnosis (2.03 +/- 0.98, P<0.05) compared to non-PD (2.04 +/-1.12, Fig 6D).

NLR, a common marker of stress and inflammation, was significantly elevated -10 to -5 years (2.72 +/- 1.32, P<0.001), 0–5 years before diagnosis (2.93 +/- 1.31, P<0.001) and 0–5 years after diagnosis in males (2.84 +/- 1.34, P<0.01) compared to non-PD (2.48 +/- 1.25, Fig 6E), with a similar pattern in females; -10 to -5 years (2.37 +/- 2.10, P<0.001), 0–5 years before diagnosis (2.47 +/- 1.11, P<0.001) and 0–5 years after diagnosis (2.96 +/- 1.57, P<0.001 and non-PD (2.19 +/- 1.08, Fig 6F). The relative risk of PD increases significantly with an elevated NLR, but that risk is reduced when you compare those that were taking ibuprofen at baseline to those that weren't (Fig 7). Whilst ibuprofen shows a strong protective effect at all NLRs, it is most apparent in those with the highest NLR ratio, suggesting inflammation plays an important role in the disease mechanism.

According to the IDEARS model C-reactive protein, demonstrated a protective relationship with PD with a relatively high SHAP score (8th in males, 16th in females); however, no time-points significantly decreased compared to non-PD in either sex (Fig 6G & 6H).

In summary, inflammatory features in males were most consistently different to non-PD at -10 to -5 years diagnosis and after diagnosis. In contrast, the only predictive inflammatory biomarker in females was the NLR which was increased at -10 to -5 years before, and 0 to 5 years after diagnosis. NLR appears to be the most consistent inflammatory biomarker associated with PD, and ibuprofen consumption appears to mitigate the negative effects of an elevated an NLR ratio.

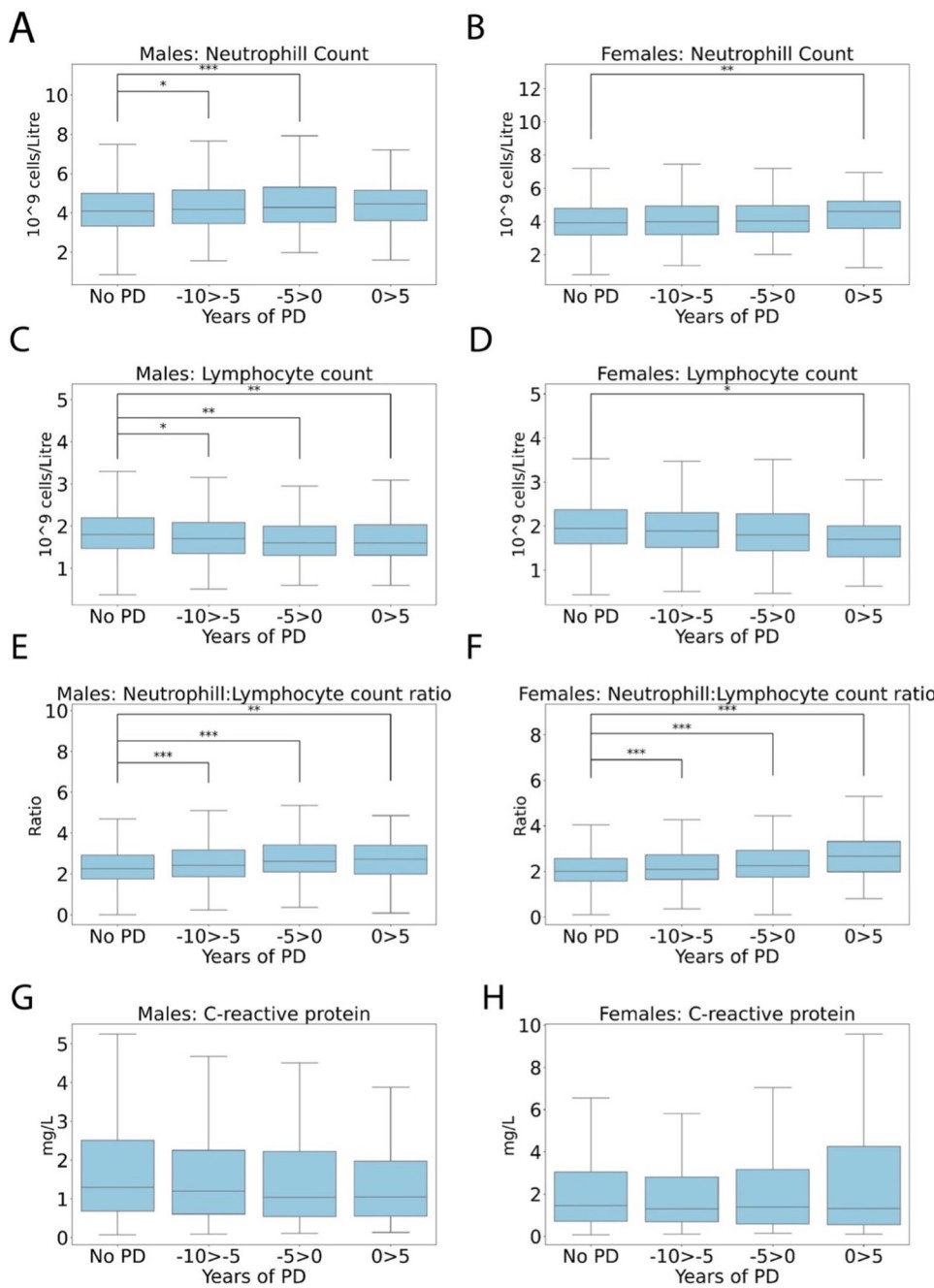

**Fig 6.** Box plots showing blood inflammatory markers for (**A, C, E, G**) males and females (**B, D, F, H**) in the 10 years preceding and 5 after a PD diagnosis in 2,036 males and 1,242 females compared to the non-PD group. Mean +/- SD. An unpaired 2-sample t-test was used for statistical comparisons, with Benjami-Hochberg correction for multiple comparisons used to calculate the adjusted p values, *P<0.05, **P<0.01 and ***P<0.001.

## Frailty

The IDEARS model revealed overall health rating, usual walking pace and total treatment/ medications as being highly associated with the development of PD. Therefore we investigated several frailty-related variables in the years preceding and following diagnosis (Fig 8). Total ICD10 diagnoses were significantly increased at -10 to -5 years (3.47 +/- 4.66, male, p<0.001;

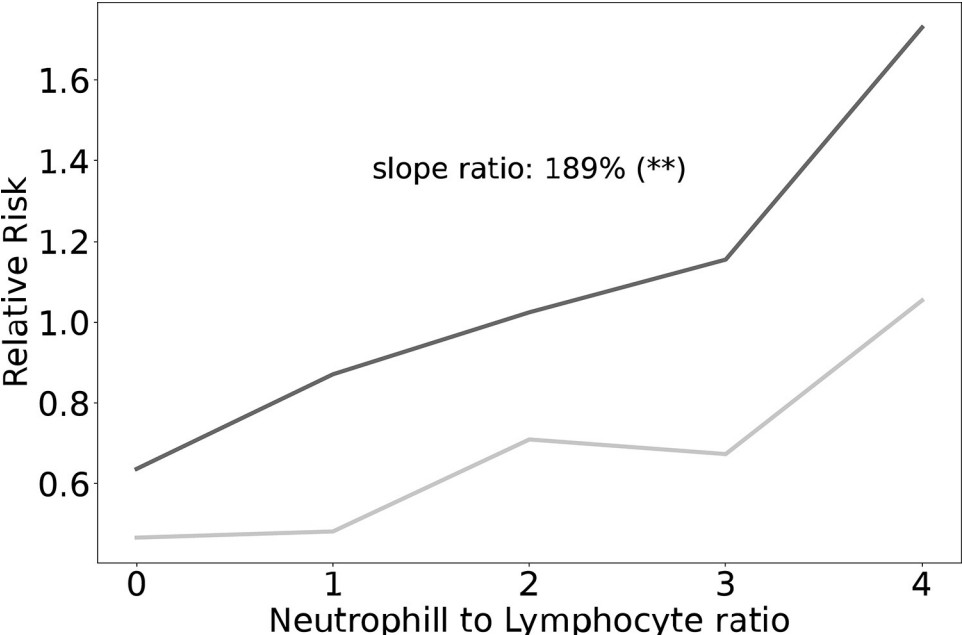

**Fig 7. Line graph showing the relationship between the relative risk of being diagnosed with PD and the neutrophil: Lymphocyte ratio (NLR) quintiles, between those taking the non-steroidal anti-inflammatory, ibuprofen at baseline.** Black line—no ibuprofen, grey line–ibuprofen. A test for equality of regression coefficients was used for statistical comparisons, *P<0.05, **P<0.01 and ***P<0.001.

3.51 +/- 4.96, female, p<0.001), -5 to 0 years (3.70 +/- 4.83, male, p<0.001; 4.32 +/- 5.31, female, p<0.001) and 0–5 years after diagnosis (7.24 +/- 6.77, male, p<0.001; 7.56 +/- 6.48, female, p<0.001) compared to non-PD (2.47 +/- 3.81, male; 2.65 +/- 3.90, female, all comparisons P<0.001, Fig 8A & 8B).

Total treatment/medications was similarly correlated with a greater risk of PD, being significantly increased at -10 to -5 years (3.18 +/- 2.87, male; 3.38 +/- 2.91, female), -5 to 0 years (4.22 +/- 3.16, male; 4.67 +/- 3.24, female) and 0–5 years after diagnosis (5.56 +/- 3.62, male; 6.31 +/- 3.29, female) compared to non-PD (2.45 +/- 2.62, male; 2.74 +/- 2.75, female, all comparisons P<0.001, Fig 8C & 8D).

Considering that PD is a movement disorder, it was unsurprising that grip strength in both hands (Fig 8E–8H), and usual walking pace were significantly reduced for both sexes at all timepoints (P<0.001 for all comparisons). Perhaps the most interesting observation is that these decreases are apparent at up to 10 years before diagnosis occurs.

Greater forced vital capacity was demonstrated be protective particularly in females, however a significant reduction was seen in both sexes. Forced vital capacity was significantly reduced at -10 to -5 years in females (2.91 +/- 0.61, p<0.001), -5 to 0 years (4.16 +/- 0.86, male, p<0.05; 2.88 +/- 0.73, female, p<0.01) and 0–5 years after diagnosis (2.81 +/- 0.68, female, p<0.001) compared to non-PD (4.29 +/- 0.89, male; 3.03 +/- 0.67, female, Fig 8I & 8IJ). Overall, these frailty-related features show very strong associations with PD risk in both sexes even from 10 years before diagnosis.

## Cardiovascular features and body adiposity

Fig 9 shows a range of cardiovascular features at timepoints before and after PD diagnosis. Decreased total cholesterol correlated with a lower risk of PD in both sexes, cholesterol was

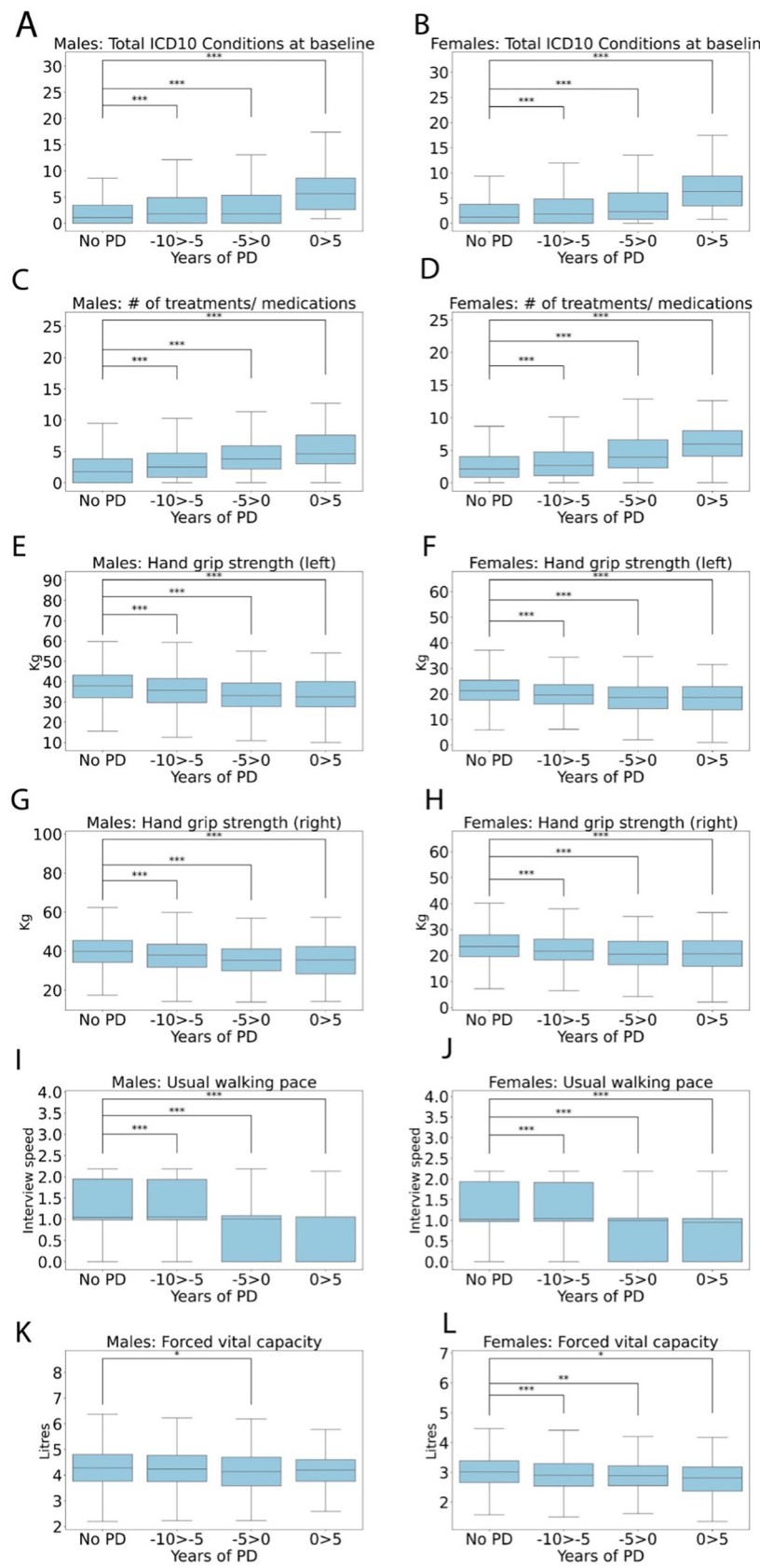

**Fig 8.** Box plots showing frailty-related variables for (**A, C, E, G, I, K**) males and females (**B, D, F, H, J, L**) in the 10 years preceding and 5 after a PD diagnosis in 2,036 males and 1,242 females compared to the non-PD group. Mean +/-SD. An unpaired 2-sample t-test was used for statistical comparisons, with Benjami-Hochberg correction for multiple comparisons used to calculate the adjusted p values, *P<0.05, **P<0.01 and ***P<0.001.

significantly reduced at both -10 to -5 years (5.19 +/- 1.11, P<0.001), -5 to 0 years (5.12 +/- 1.06, P<0.001) and 0–5 years after diagnosis in males (5.01 +/- 1.15, P<0.001) compared to non-PD (5.47 +/- 1.13, Fig 9A). In females, cholesterol was significantly decreased compared to non-PD (6.04 +/- 1.13) at -5 to 0 years (5.77 +/- 1.15, P<0.001). Given the protective effect of total cholesterol we investigated HDL and LDL cholesterol. For HDL there was a significant reduction at -5 to 0 years (1.55 +/- 0.40, P<0.05) in females compared to non-PD (1.62 +/- 0.38), with all other levels consistent across groups (Fig 9C & 9D). A reduction in LDL was apparent a multiple timepoints for both sexes. LDL was significantly reduced at -10 to 5 years (3.26 +/- 0.85, P<0.001) and -5 to 0 years before diagnosis (3.20 +/- 0.83, P<0.001) and 0–5 years after diagnosis (3.17 +/- 0.92, P<0.001) in males compared to non-PD (3.46 +/- 0.86, Fig 9E). In females, LDL was significantly decreased compared to non-PD (3.56 +/- 0.87, P<0.01) only at -5 to 0 years compared to non PD (3.74 +/- 0.88, Fig 9F). Interestingly, waist circumference was significantly increased in females compared to non-PD (85.31 +/- 12.29) at -10 to -5 years before diagnosis (86.28 +/- 12.50, female, p<0.05, Fig 9G & 9H), and trends higher at other timepoints. In summary, total cholesterol and LDL appear to be protective, this indicates people with a reduced risk of heart disease a more likely to developed PD, however increased central adiposity, indicated by increased waist circumference in females is associated with increased PD risk in the next 10 years.

## Discussion

We presented the IDEARS platform, which uses state-of-the-art machine learning algorithms XGBoost and SHAP to rank the risk factors for PD using the world's largest and most comprehensive prospective community study, the UK Biobank. Ageing is widely recognised as the most significant factor in predicting PD, therefore we chose to age normalise our datasets to uncover a hierarchy of feature importance that is age-independent. Our model demonstrated that gender was the most important feature, with PD being more prevalent in males, which led us to further split subsequent analyses by gender to uncover gender specific feature importances. Our unbiased machine learning approach uncovered a novel set of features most associated with PD. Interestingly, several well-established risk factors thought to have a high association level with PD were not identified in the most important features in our model (e.g., pesticide exposure, smoking status, traumatic brain injury and caffeine consumption).

Of note is the importance of insulin-like growth factor 1 (IGF-1), which presented in the top 3 most important features, based on mean SHAP score in the combined dataset, and male and female lists. On deeper inspection of the data, it was clear that IGF-1 levels were elevated in males and females up to 10 years before disease onset. IGF-1 is an endocrine, paracrine and autocrine hormone that is a primary mediator of the effects of growth hormone. Major functions of IGF- include insulin-like activity, cell proliferation and survival, antioxidant effects and neuroprotection. *In vivo* studies have demonstrated IGF-1 deficiency results in increased oxidative stress, inflammation, neuronal cell death and cognitive deficits that can be improved by exogenous IGF-1 [19, 20]. It is well documented that IGF-1 is elevated in serum at diagnosis in PD patients, and levels at this time correlate with disease severity [5, 7]. To account for the discrepancy in the beneficial effects of IGF-1 and the fact it is increased in PD, it has been hypothesised that IGF-1 signalling is defective in PD, resulting in a decrease in the

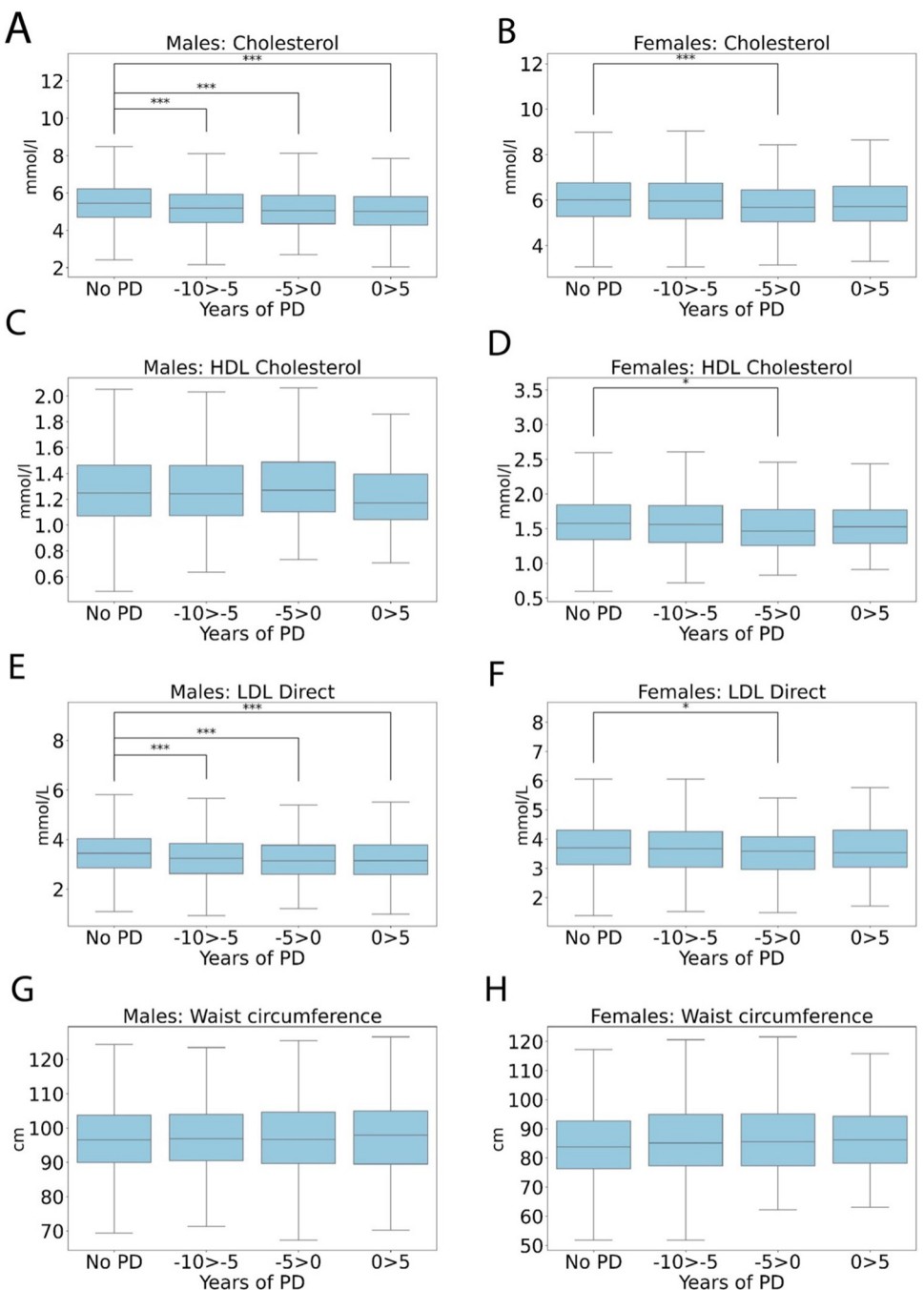

**Fig 9.** Box plots showing cardiovascular variables and waist circumference for (**A, C, E, G**) males and females (**B, D, F, H**) in the 10 years preceding and 5 after a PD diagnosis in 2,036 males and 1,242 females compared to the non-PD group. Mean +/- SD. An unpaired 2-sample t-testwas used for statistical comparisons, with Benjami-Hochberg correction for multiple comparisons used to calculate the adjusted p values, *P<0.05, **P<0.01 and ***P<0.001.

neuroprotective effects and reduction in the brains ability to buffer oxidative damage. Moreover, IGF-1 signalling is known to be dysregulated by both toxin-induced inflammation and central obesity [5, 21, 22], consistent with our model identifying prospective biomarkers predictive of greater PD risk in these categories. Therefore higher-than-average IGF-1 levels years

before diagnosis may indicate a compensatory mechanism in response to dysregulated IGF-1 signalling. Our findings suggest that IGF-1 should be further considered as a prognostic biomarker for PD risk.

AST:ALT was elevated up to 10 years before, and after PD diagnosis in males but not in females, this is consistent with elevated ALT being protective in the male SHAP list. Elevated AST:ALT ratios between 1–2 are indicative of non-alcoholic fatty liver disease (NAFLD) or non-alcoholic steatohepatitis (NASH), whilst levels <2 are indicative of alcoholic liver disease [23, 24], therefore the moderate increases in the male UKB PD cohort may be indicative of NAFLD/NASH, although some individuals in the PD group have levels above 2. A recent study of NAFLD and PD found a greater risk of PD in females with NAFLD [25], and an earlier study found that NASH in males and females with hepatitis B and C infection led to a greater PD risk [26]. With that said, NAFLD is associated with cardiovascular disease and metabolic disorders which does not fully align with our other findings (see below) [27]. Whilst more research on NAFLD and PD is required, our findings indicate elevated AST:ALT may be a useful prospective biomarker of PD in males.

The IDEARS model identified several features associated with cardiovascular health and body adiposity. Total and LDL cholesterol levels were reduced in PD in males 10 years before diagnosis but only 5 years in females. This observation is in keeping with a large population-based study of 261,638 statin-free individuals, which identified that males who had lower levels of total and LDL cholesterol were at a greater risk of developing PD, however there was no significant differences in females [8]. Given lower LDL levels, PD patients have shown a reduced risk of myocardial infarction and stroke [28, 29]. It has been hypothesised that the reduced cholesterol levels may be due to nonmotor peripheral symptoms, such as constipation, that can manifest before motor symptoms appear [8].

Cardiovascular health is also strongly linked to metabolic regulation. There are mixed findings on the co-morbidity of type 2 diabetes and PD, with some studies showing an increase [30], and others showing a reduced prevalence [29, 31]. As mentioned above HbA1c is higher 10 years before PD onset in males, but reduced 0–5 years before diagnosis. However, the proportion of the PD group with HbA1c in the diabetic range is slightly higher than the non-PD group. Therefore, further research is needed to investigate the possible associations of diabetes and PD.

In keeping with previous literature, the IDEARS platform identified that increasing urate concentrations are associated with a lower risk of PD [32, 33]. It is thought that urate reduces the risk of neurodegenerative diseases through its iron chelating properties, antioxidant quenching of superoxide and hydroxyl free radicals, and as an electron donor that increases antioxidant activity of enzymes, such as superoxide dismutase [34]. IDEARS identified increased creatinine levels in the urine of both sexes before and after PD diagnosis, which may be indicative of poor kidney function beginning in the pre-symptomatic phase. However this finding is at odds with a large Swedish study that found a slight reduction in creatinine serum levels from 1 year before diagnosis onwards [35], whilst another smaller study found no change in serum creatinine in PD [36]. Therefore, whilst decreased urate levels might be a useful biomarker for PD, further investigations are required to understand the relationship of creatinine and PD.

Several epidemiological studies have linked central adiposity to PD [37, 38], consistent with output from the IDEARS model with waist circumference being ranked 14[th] and significantly increasing before diagnosis in females. Although this observation may be at odds with better cardiovascular and metabolic health in general, body fat distribution may be a key factor, with increased adiposity hypothesised to modulate IGF-1 signalling [5, 21]. Clearly, more research is required to better understand the complex interactions of body adiposity and the risk of PD.

The IDEARS model identified several features relating to the immune system, specifically an increase in neutrophil count, a decrease in lymphocyte count and an increase in NLR, were all identified to be altered both 10 years before and at diagnosis in males, whilst only NLR followed the same pattern in females. An elevated neutrophil count is associated with the occurrence, progression, and severity of inflammation or infection. In contrast, a decreased lymphocyte count, as part of the adaptive immune response, is heavily depressed by stress. Thus, NLR is considered a compound biomarker of inflammation and stress. Therefore it is perhaps not surprisingly that NLR is the most robust and consistent example of a prospective biomarker of PD risk from the IDEARS model. A recent study demonstrated similar findings with increased NLR in 100 PD patients, but no change in Alzheimer's disease [6]. Increased neutrophil count and NLR are in keeping with the literature that inflammation and infection are risk factors for PD. NLR may therefore be considered a useful prospective biomarker for the risk of PD, however as it is associated with many other chronic diseases, it should be used in combination with other biomarkers identified by our model.

Unexpectedly, C-reactive protein, a marker of acute inflammation, appeared as protective in the SHAP list, however on closer inspection it was unchanged in the PD cohort before or after diagnosis in both sexes. The appearance of C-reactive protein as protective, may be due to its complex relationships with other inflammatory markers, which may be more chronic. No change in C-reactive protein is at odds with a recent meta-analysis that found an increase in C-reactive protein in PD [39]. This may be explained as the UKB dataset includes 2,719 subjects with PD, compared to a combined 2,691 subjects across twenty studies in the meta-analysis. Moreover, findings in those studies were highly variable, two showed no change in C-reactive protein, 10 a small increase and 8 showed a large increase. Therefore, the usefulness of C-reactive protein as a biomarker of PD remains an open question.

Epidemiological studies have revealed viral (e.g. *influenza*, *HSV*, *hepatitis*) and bacterial (e.g. *C. pneumonia* and *H. pylori*) infections are associated with an increased risk of developing PD [26, 40–43]. Inflammatory conditions, such as head trauma, allergic rhinitis and exaggerated allergic reactions following insect stings, have been linked to an increased risk of developing PD [44–47]. Neuroinflammation is also a common pathological hallmark seen in the PD brain [48–51]. Conversely, long-term use of non-steroidal anti-inflammatory drugs (NSAIDs) reduce the risk of developing PD [52–55]. Our analysis demonstrates a protective effect of Ibuprofen use in the UKB participants, which was more pronounced in at higher NLR.

The reduction in lymphocyte count well before PD in our study is consistent two recent studies, including one that used the UKB dataset (thus validating our approach) [11, 32], as well as a meta-analysis that showed decreased numbers of CD3+ and CD4+ lymphocyte subsets in intermediate and late-stage PD, whilst a decrease in CD8+ T lymphocytes was also observed [56]. Interestingly, this reduction in lymphocyte count occurs up to 10 years before diagnosis in males, but only 5 years before in females, and therefore maybe a better prospective marker in men. It is noteworthy that 'suffers from nerves' (19th overall) and self-reported nervous feeling (8th in females) were highly ranked risk factor in the IDEARS model. Therefore the PD group may have higher-than-average stress levels, which could depress lymphocyte counts. More detailed analyses of CD4+ T lymphocyte subsets suggests that they are skewed towards proinflammatory phenotypes (i.e., increased Th1, Th17, and reduced Th2 and Tregs) in PD patients [57–59]. The inflammatory milieu in PD may also be a contributor to decreased IGF-1 signalling mentioned previously [5, 19, 20]. Overall, these findings imply a predisposition to PD may be established by conditions that induce peripheral inflammation (injury/infection) and stress, or in individuals with an immune system skewed towards inflammation.

Given that PD is an age-related motor disease it was unsurprising that the IDEARS model identified overall health rating and number of treatments/medications taken as highly ranked

features, indicative of overall frailty. A deeper analysis into other frailty related features revealed reduced hand grip strength and decreased walking pace can be considered early markers of motor dysfunction and given they are significantly reduced in both sexes 10 years before diagnosis they should be considered as useful clinical measures to predict the risk of PD onset. Existing literature has identified the importance of these factors. Hand grip strength and reduced dexterity have been reported as predictors of motor symptom severity in PD [60]. Slow walking speed has been correlated with advanced age and PD severity [61], and it is also one of the first complaints in the early stages of the disease [62]. Increased number of ICD conditions at baseline, and reduced forced vital capacity were also apparent years before diagnosis in both sexes, and are indictors of general ill health and multiple co-morbidities in PD patients. Arthritis, hypertension, atrial fibrillation, depression, back problems, and cataracts are commonly reported co-morbidities of PD [30, 63], and require a wide range of treatments.

Other significant gender differences were observed with parental PD being more important for men than women, which may suggest that since idiopathic PD has a phenotype that strongly overlaps with monogenic forms of the disease [64], there may be a greater genetic component in idiopathic PD in males. Conversely, vitamin D was more protective for PD in women than men. Vitamin deficiency has been linked to neurodegenerative diseases, and a deficiency in vitamin D in particular has been linked to reduced dopamine levels and alpha-synuclein accumulation, which are pathological hallmarks of PD [65]. Vitamin D has been shown to have neuroprotective, anti-inflammatory and antioxidant effects *in vitro* [66], however a recent meta-analysis could not conclude clear benefits of vitamin D supplementation in reducing PD risk [67].

Applying a novel methodology in the IDEARs platform has enabled us to examine a much larger range of variables without *a priori* assumption. The advantage of using XGBoost and SHAP in this context is the ability to consider a large number of variables and accurately determine their importance in the model while implicitly modelling interactions between variables, resulting in a demonstratively higher AUC. The disadvantage is the black box nature of this approach. We have sought to mitigate this by providing a separate univariate analysis of individual variables. It is important to state that a limitation to this modelling approach is that it does not imply causality, Bayesian Networks [68] and other approaches seek to better understand the direction of causality between factors and would be a natural extension to this work. In addition to the power of determining the most significant risk factors in driving PD, this approach could be used separately to provide a risk score which we would expect to be more accurate than existing methods.

## Conclusion

In summary our novel unbiased model "IDEARS" identified a novel set of risk factors for PD that diverge considerably from the most well-established risk factors thought to have a high association level with PD. The most promising biomarkers for PD risk are elevated IGF-1, AST:ALT, NLR and reduced urate, and total and LDL cholesterol. These biomarkers demonstrated a consistent change before PD onset in males, however only IGF-1 and NLR were robustly elevated before diagnosis in females. Given the non-specific nature of some of these biomarkers (e.g. AST:ALT, NLR), we suggest that they would be best used in combination to predict PD risk. If the hoped-for development of neuroprotective treatments for PD is fruitful, our biomarker panel may help identify those at heightened risk who may benefit most from prophylactic treatment. Features indicative of frailty, particularly those that relate to motor dysfunction, such as walking pace and hand grip strength, as well as a high number of co-

morbidities and poor overall health rating, were strongly associated with increased PD risk in both sexes, and these signs could serve as useful clinical indications leading to earlier diagnosis.

## Supporting information

**S1 File.**
(XLSX)

## Author Contributions

**Conceptualization:** Michael Allwright, Hamish Mundell, Greg Sutherland, Paul Austin, Boris Guennewig.

**Data curation:** Michael Allwright, Hamish Mundell.

**Formal analysis:** Michael Allwright.

**Methodology:** Michael Allwright.

**Software:** Michael Allwright.

**Supervision:** Paul Austin, Boris Guennewig.

**Validation:** Greg Sutherland.

**Visualization:** Michael Allwright.

**Writing – original draft:** Michael Allwright, Paul Austin.

**Writing – review & editing:** Michael Allwright, Greg Sutherland, Paul Austin, Boris Guennewig.

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
