## [Decision Letter · Decision Letter 0]

24 Jan 2023

PONE-D-22-27269Machine Learning Analysis of the UK Biobank Reveals IGF-1 and Inflammatory Biomarkers Predict Parkinson’s Disease RiskPLOS ONE

Dear Dr. Allwright,

Thank you for submitting your manuscript to PLOS ONE. After careful consideration, we feel that it has merit but does not fully meet PLOS ONE’s publication criteria as it currently stands. Therefore, we invite you to submit a revised version of the manuscript that addresses the points raised during the review process.

 Please submit your revised manuscript by Mar 10 2023 11:59PM. If you will need more time than this to complete your revisions, please reply to this message or contact the journal office at plosone@plos.org. Please include the following items when submitting your revised manuscript:A rebuttal letter that responds to each point raised by the academic editor and reviewer(s). You should upload this letter as a separate file labeled 'Response to Reviewers'.A marked-up copy of your manuscript that highlights changes made to the original version. You should upload this as a separate file labeled 'Revised Manuscript with Track Changes'.An unmarked version of your revised paper without tracked changes. You should upload this as a separate file labeled 'Manuscript'.

We look forward to receiving your revised manuscript.

Kind regards,

Jacopo Sabbatinelli, MD, PhD

Academic Editor

PLOS ONE

Journal Requirements:

Reviewers' comments:

Reviewer's Responses to Questions

**Comments to the Author**

1. Is the manuscript technically sound, and do the data support the conclusions?

Reviewer #1: Yes

2. Has the statistical analysis been performed appropriately and rigorously? 

Reviewer #1: I Don't Know

3. Have the authors made all data underlying the findings in their manuscript fully available?

Reviewer #1: No

4. Is the manuscript presented in an intelligible fashion and written in standard English?

Reviewer #1: Yes

5. Review Comments to the Author

Reviewer #1: 1. The authors should have included the details of several analysis methods in the methodology part of the manuscript, which are essential for the analysis evaluation.

1.1 Please provides hyperparameter tuning methods.

1.2 Please specify the AUC means comparison methods in figure 3.

1.3 Please specify the statistical methods used in figures 5, 6, 7, 8, and 9.

2. After imputation, data duplication-checking might be needed.

3. Please provide the definition of "overall health rating". Is this feature independent from other features?

4. What is the meaning of "the unbiased IDEARS model"? What is the difference from the IDEARS model? How to define the unbiased property.

5. Is it possible to provide the list of features; Va, and Vc?

6. Is it possible to provide the code of analysis pipeline?

7. The authors should have included proper references to the methods and libraries; XGBoost, and SHAP; used in the IDEARs platform.

6. PLOS authors have the option to publish the peer review history of their article (what does this mean?). If published, this will include your full peer review and any attached files.

Reviewer #1: No

---

## [Author Response · Author response to Decision Letter 0]

20 Feb 2023

1.1 Please provides hyperparameter tuning methods.

Additional details of the hyper-parameter tuning method used have been added to the revised manuscript.

1.2 Please specify the AUC means comparison methods in figure 3.

Details of the statistical comparison of the AUC means between each model have been added to the methods. The statistical test used, and the result of the AUC means comparison has also been added to the figure legend for Figure 3.

1.3 Please specify the statistical methods used in figures 5, 6, 7, 8, and 9.

This information is contained in the methods:

“For both the male and female cohorts unpaired 2-sample t-tests were performed to compare the means for each variable in the non-PD group with the means of each variable at each disease stage: 5-10 years prior to disease diagnosis, 0-5 years prior to disease diagnosis and 0-5 years post disease diagnosis respectively.”

Also the statistical test used and the significance levels have been added to each figure legend.

In addition, the reporting of the statistics in the results have been simplified to report, P<0.05, P<0.01 or P<0.001 rather than specific p values, and some errors have been corrected, e.g. greater than sign was incorrectly used instead of less than sign.

2. After imputation, data duplication-checking might be needed.

Thank you for raising this concern. Data duplication checking was performed as one of the checks to confirm the pipeline was working as planned. We confirm that the datasets used post imputation contained no duplicate values. We had already pre-selected variables which were populated for >80% of participants. Also, our section on stratification by gender and disease progression (Figures 5,6,8 and 9) used non-imputed data and excluded null values from the analysis. 

3. Please provide the definition of "overall health rating". Is this feature independent from other features?

This is a UKB questionnaire field (Data-Field 2178) and is self-reported by the participant on a 4-point scale from “poor” to “excellent”. It is independent of other features.

4. What is the meaning of "the unbiased IDEARS model"? What is the difference from the IDEARS model? How to define the unbiased property.

There is no difference between the two, and we apologise for this error. We have revised the wording in the manuscript to reflect this. We have also spelt out IDEARs as The Integrated Disease Explanation and Risk Scoring platform, an omission in the previous manuscript, so we hope all is clear now. 

5. Is it possible to provide the list of features; Va, and Vc?

The list of features has been provided as a supplementary file supp_file1.xlsx.

6. Is it possible to provide the code of analysis pipeline?

The location code of the analysis pipeline is provided in the methods, and can be found at this link: https://github.com/binfnstats/ukb-IDEARS.

7. The authors should have included proper references to the methods and libraries; XGBoost, and SHAP; used in the IDEARs platform.

Thanks for raising this and we apologise for this oversight. We have now referenced the original XGBoost paper. The SHAP reference is to https://arxiv.org/abs/1705.07874 which we understand to be the correct paper to cite based on this advice https://github.com/slundberg/shap, as we use SHAP in a “general sense”. We have added a citation to this paper to the Methods section.

---

## [Decision Letter · Decision Letter 1]

6 Mar 2023

PONE-D-22-27269R1Machine Learning Analysis of the UK Biobank Reveals IGF-1 and Inflammatory Biomarkers Predict Parkinson’s Disease RiskPLOS ONE

Dear Dr. Allwright,

Thank you for submitting your manuscript to PLOS ONE. After careful consideration, we feel that it has merit but does not fully meet PLOS ONE’s publication criteria as it currently stands. Therefore, we invite you to submit a revised version of the manuscript that addresses the points raised during the review process.

We look forward to receiving your revised manuscript.

Kind regards,

Jacopo Sabbatinelli, MD, PhD

Academic Editor

PLOS ONE

Journal Requirements:

Reviewers' comments:

Reviewer's Responses to Questions

**Comments to the Author**

1. If the authors have adequately addressed your comments raised in a previous round of review and you feel that this manuscript is now acceptable for publication, you may indicate that here to bypass the “Comments to the Author” section, enter your conflict of interest statement in the “Confidential to Editor” section, and submit your "Accept" recommendation.

Reviewer #1: All comments have been addressed

2. Is the manuscript technically sound, and do the data support the conclusions?

Reviewer #1: Yes

3. Has the statistical analysis been performed appropriately and rigorously? 

Reviewer #1: No

4. Have the authors made all data underlying the findings in their manuscript fully available?

Reviewer #1: Yes

5. Is the manuscript presented in an intelligible fashion and written in standard English?

Reviewer #1: Yes

6. Review Comments to the Author

Reviewer #1: Thank you so much for your responses.

According to the response, the statistical methods used in figures 5, 6, 7, 8, and 9 was unpaired 2-sample t-tests. The results show that the comparison is between 3 groups of PD patients and non-PD and involved several independent variables, which is multiple testing. The unpaired 2-sample t-tests might not be appropriate for the analysis.

7. PLOS authors have the option to publish the peer review history of their article (what does this mean?). If published, this will include your full peer review and any attached files.

Reviewer #1: No

---

## [Author Response · Author response to Decision Letter 1]

3 Apr 2023

Responses to Reviewers

3. Has the statistical analysis been performed appropriately and rigorously?

Reviewer #1: No

Response: Thank you for pointing out the need for us to make a correction given the multiple comparisons, please see below for our full response here. 

6. Review Comments to the Author

Reviewer #1: Thank you so much for your responses.

According to the response, the statistical methods used in figures 5, 6, 7, 8, and 9 was unpaired 2-sample t-tests. The results show that the comparison is between 3 groups of PD patients and non-PD and involved several independent variables, which is multiple testing. The unpaired 2-sample t-tests might not be appropriate for the analysis.

Response: Thank you for pointing this out and we apologise for failing to apply this originally. Due to your insights here, we have been able to make these changes using the “Benjami-Hochberg correction for multiple comparisons” (reference 18 in revised paper). This has reduced the associated p values and hence the statistical significance of certain variables, but it has not affected the key conclusions of the paper. We feel that thanks to this addition, this section is now more rigorous.

---

## [Editor Report · Decision Letter 2]

24 Apr 2023

Machine Learning Analysis of the UK Biobank Reveals IGF-1 and Inflammatory Biomarkers Predict Parkinson’s Disease Risk

PONE-D-22-27269R2

Dear Dr. Allwright,

We’re pleased to inform you that your manuscript has been judged scientifically suitable for publication and will be formally accepted for publication once it meets all outstanding technical requirements.

Kind regards,

Jacopo Sabbatinelli, MD, PhD

Academic Editor

PLOS ONE
---

## [Editor Report · Acceptance letter]

28 Apr 2023

PONE-D-22-27269R2 

Machine Learning Analysis of the UK Biobank Reveals IGF-1 and Inflammatory Biomarkers Predict Parkinson’s Disease Risk 

Dear Dr. Allwright:

I'm pleased to inform you that your manuscript has been deemed suitable for publication in PLOS ONE. Congratulations! Your manuscript is now with our production department. 

Kind regards, 

on behalf of

Dr. Jacopo Sabbatinelli 

Academic Editor

PLOS ONE